# Development Modeling of *Phormia regina* (Diptera: Calliphoridae)

**DOI:** 10.3390/insects15070550

**Published:** 2024-07-21

**Authors:** Amanda Roe, Jeffrey D. Wells, Leon G. Higley

**Affiliations:** 1Biology Program, College of Saint Mary, Omaha, NE 68583, USA; aroe@csm.edu; 2Department of Biological Sciences, Florida International University, Miami, FL 33199, USA; jedwell@fiu.edu; 3School of Natural Resources, University of Nebraska-Lincoln, Lincoln, NE 68583, USA

**Keywords:** postmortem interval, forensic entomology, carrion, temperature, blow fly

## Abstract

**Simple Summary:**

Simple Summary: The blow fly *Phomia regina* is arguably the most important forensic indicator species in the United States of America. Typically, the development rate of forensically important insects is used to estimate the postmortem interval in situations with decomposing bodies. Consequently, this study examined the development of *P. regina* across 11 temperatures. Evidence demonstrated that maggots have a normal distribution in their transitions between stages. The large data set allowed for the development of a comprehensive degree-day model across all life stages.

**Abstract:**

A series of experiments were conducted on *Phormia regina*, a forensically important blow fly species, that met the requirements needed to create statistically valid development models. Experiments were conducted over 11 temperatures (7.5 to 32.5 °C, at 2.5 °C intervals) with a 16:8 L:D cycle. Experimental units contained 20 eggs, 10 g of beef liver, and 2.5 cm of sand. Each life stage (egg to adult) had five sampling times. Each sampling time was replicated four times for a total of 20 measurements per life stage. For each sampling time, the cups were pulled from the chambers, and the stage of each maggot was documented morphologically through posterior spiracular slits and cephalopharyngeal development. Data were normally distributed with the later larval (L3m) and pupation stages having the most variation within and transitioning between stages, particularly between 12.5 °C and 20.0 °C. The biological minimum was between 10.0 °C and 12.5 °C, with little egg development and no egg emergence at 7.5 °C and no maturation past L1 at 10.0 °C. *Phormia regina* did not display increased mortality associated with the upper temperature of 32.5 °C. The development data generated illustrate the advantages of large data sets in modeling blow fly development and the need for curvilinear models in describing development at environmental temperatures near the biological minima and maxima.

## 1. Introduction

The most common forensic entomological analysis includes inferring the age of a carrion insect, e.g., a blow fly maggot. Under typical circumstances, this value can be interpreted as a minimum postmortem interval (PMI_min_) [1]. Specimen age is usually determined by comparison to laboratory development rate data [2]. Insect development rate is profoundly influenced by environmental temperature, and the effect of temperature on carrion insect development has been extensively investigated [3]. However, it is extremely unlikely that laboratory development has been observed at the same temperatures as the corpse. Therefore, the forensic entomologist may use some method to extrapolate from the rate of development observed at one (or more) temperature(s) to predict the development rate at the corpse temperatures. One generally accepted method for this is to assume that the development rate is approximately a linear function of temperature. To the extent that this is true (the linear portion of the temperature–development curve), the time required to complete a given amount of development, e.g., oviposition to pupation, will require the same number of accumulated degree hours (ADHs) at any temperature history [4].

Calculating age in ADHs requires a minimum development temperature [3]. For linear developmental models, the minimum developmental temperature corresponds to the x-intercept of a temperature–developmental rate linear regression. The minimum developmental temperature is different from the biological minimum, which is the temperature below which development does not proceed. Such models require even temperature intervals that span the entire linear portion of the temperature developmental curve [3]. Ideally, these should include multiple points defining stage transitions [5].

In the context of a death investigation, an estimate of insect age should be a range, defined by an explicit method acceptable to the statistical science community, and associated with a probability. One such approach for estimating age from developmental stage was described by Wells and LaMotte [6]. This requires suitable estimates of ADH requirements for each developmental stage and a representative sample size that depends on the number of life stages to be distinguished (e.g., >37 for six different in-stars) [7]. One important fly lacking the above is *Phormia regina*.

The blow fly *P. regina* is among that group of necrophagous insects that require decomposing tissue to complete their life cycle. They are Holarctic in distribution and are very common throughout the U.S., except in southern Florida [8,9,10]. While *P. regina* is considered a “cold weather fly” and moves north as temperatures increase, they can be found in large numbers throughout the central U.S. during the hottest months of the year and are typically in much higher numbers than other blow fly species (Roe, personal obs.). *Phormia regina*’s ability to quickly colonize an area and maintain large populations means that it has quickly become one of the most common blow fly species recovered at human and other animal death scenes, making them incredibly important in postmortem interval (PMI) estimations.

*Phormia regina* development has been investigated by multiple researchers [8,11,12,13,14,15,16], Table 1). Unfortunately, methodological problems exist in some instances: insufficient replication or data points, inconsistent temperature ranges or too few temperatures, non-life stage specific results, and an inability to apply error rates or confidence intervals. These differences make it difficult to compare and/or pool data for analysis.

The issues related to differing methodologies and analyses were discussed at length in [14,17], with the overall conclusion that the accuracy of PMI estimations can be directly related to sampling errors and too few temperatures studied. Although it is important to be able to compare data sets and have them come from similarly conducted experiments, a more applicable reason for consistent data is that known standards and error rates are required under the Daubert standards and the Federal Rules of Evidence when testifying in a court of law. Judges can use “the known or potential rate of error of the technique or theory when applied” as an assessment of the reliability of the evidence presented.

A series of experiments were conducted that covered a large range of temperatures, had large sample sizes, and had consistent sampling times, which are required to create statistically valid development models.

## 2. Methods and Materials

The first three sections of Materials and Methods presented here are essentially the same as in [4]. They have been repeated here for ease of reading and understanding.

### 2.1. Flies

*Phormia regina* was obtained from colonies maintained at the University of Nebraska-Lincoln (Lincoln, Nebraska). The colony was established in 2011 from field-collected insects in Lancaster County, Nebraska. At the research time (2013–2014), the colonies had achieved 100 generations without the addition of new flies to reduce genetic variation within the colony. The intent was to “obtain genetic homogeneity among test subjects, so we can get an indication of physiological variation in response without confounding from population variation. Thus, the results here are intended as a baseline against which potential variation among populations can be tested. The chief danger in using such inbred lines experimentally is the potential for inadvertent selection. With insects, inadvertent selection in colonies most frequently occurs in oviposition behavior and in reduced fecundity; however, no indications of change in either of these factors were observed in any of our colonies over many generations” [18]. Adult flies were maintained in screen cages (46 cm × 46 cm × 46 cm) (Bioquip Products, Compton, CA, USA) in a rearing room at 27.5 °C (±3 °C), with a 16:8 (L:D) photoperiod. Multiple generations were maintained in a single cage, and ca. 1000 adult flies were introduced every 1–2 weeks. Adults had access to granulated sugar and water ad libitum, as well as raw beef liver for protein and as an ovipositional substrate. After egg laying, eggs and liver were placed in an 89 mL plastic cup, which was surrounded by pine shavings in a 1.7 L plastic box. The pine shavings served as a pupation substrate. The 1.7 L box was placed in an I30-BLL Percival biological incubator (Percival Scientific, Inc., Perry, IA, USA) set at 26 °C (±1.5 °C). After eclosion, adults were released into the screened cages.

### 2.2. Incubators

Incubator information has been previously discussed [19]. The pertinent information is revisited here. Incubators were customized model SMY04-1 DigiTherm^®^ CirKinetics Incubators (TriTech Research, Inc., Los Angeles, CA, USA). The DigiTherm^®^ CirKinetics Incubator has microprocessor-controlled temperature regulation, internal lighting, and recirculating air system (to help maintain humidity, ca. 80% RH) and uses a thermoelectric heat pump (rather than coolant and condenser as is typical with larger incubators and growth chambers). Customizations included the addition of a data port, vertical lighting (so all shelves were illuminated), and an additional internal fan. The manufacturer’s specifications indicate an operational range of 10–60 °C ± 0.1 °C. It is worth noting that a range of ±0.1 °C is an order of magnitude more precise than is possible in conventional growth chambers. Although growth chambers have been shown to display substantial differences between programmed temperatures and actual internal temperatures [18], incubators tested with internal thermocouples in a replicated study showed that internal temperatures on all shelves within incubators never varied more than 0.4 °C from the programmed temperature. Given the high level of measured accuracy with programmed temperatures, we were able to use incubators for temperature treatments, which improved our experimental efficiency and helped reduce experimental error.

### 2.3. Experimental Design

The study comprised eleven temperatures (7.5, 10, 12.5, 15, 17.5, 20, 22.5, 25, 27.5, 30, and 32.5 °C) with a light–dark cycle of 16:8. We chose not to include temperatures above 32.5 °C because these temperatures would be in the non-linear range of the temperature curve [12]. Twenty eggs (collected within 30 min of oviposition) were counted onto a moist black filter paper triangle and placed in direct contact with 10 g of beef liver in a 29.5 mL plastic cup. The cup was placed in a 7 cm × 7 cm × 10 cm plastic container that had 2.5 cm of wood shavings in the bottom. The container was then placed randomly in an incubator. Each life stage (egg–first stage, first–second stage, second–third stage, third–third migratory, third migratory–pupation, pupation–adult) was calculated using Kamal’s [10] data, which were converted to accumulated degree hours (ADH) and divided equally into five sampling times (Table 2). Each sample was replicated four times, for a total of 20 samples per life stage. During each sample time, a container was pulled from each of the four incubators, and the stage of each maggot was documented morphologically using the posterior spiracular slits and cephalopharyngeal skeleton.

During egg hatch, a larva was recorded as the first stage if they had broken the egg chorion and were actively emerging. Pharate larvae (larvae that have undergone apolysis but not ecdysis) were recorded as the earlier stage (e.g., third-stage spiracular slits can be seen beneath the current spiracular slits would be recorded as a second stage) since they had not yet molted. Pupariation started when the larva had a shortened body length and no longer projected its mouth hooks when put in the larval fixative KAAD (kerosene-acetic acid-dioxane). There were times when a larva appeared to be entering the puparium stage but would extend its body length and begin crawling if disturbed or placed in KAAD. These larvae were recorded as third migratory. All life stages were preserved in 70% ethyl alcohol. Third and third–migratory stages were fixed in KAAD for 48 h and transferred to 70% ethyl alcohol.

### 2.4. Analysis

Two regression procedures were used. First, to determine the appropriate transition distributions, we used TableCurve 2d, version 5.01 (SYSTAT Software Inc., San Jose, CA, USA, http://www.sigmaplot.com/products/tablecurve2d/tablecurve2d.php, accessed on 20 July 2024) and Prism, version 6.02 (GraphPad Software, Inc., La Jolla, CA, USA, http://www.graphpad.com/scientific-software/prism/, accessed on 20 July 2024). Here, we fit one of four functions (specifically, a regressed proportion (percentage) in stage versus time at each temperature tested). The equations used were the following:A Gaussian equation (a standard normal curve):
$$y=a\;exp\left[-\frac{1}{2}{\left(\frac{x-b}{c}\right)}^{2}\right]$$A modified Gaussian equation (a form of Gaussian curve with a plateau at 100%):
$$y=a\;exp\left[-\frac{1}{2}{\left(\frac{\left|x-b\right|}{c}\right)}^{d}\right]$$A cumulative Gaussian equation (a form of the Gaussian curve used for adults to model a sigmoidal increase to a plateau):
$$y=\frac{a}{2}\left[1+erf\left(\frac{x-b}{\sqrt{2}c}\right)\right]$$A reversed cumulative Gaussian equation (a form of the cumulative Gaussian equation used for eggs to model a sigmoidal decrease from a plateau):
$$y=\frac{a}{2}\left[1-erf\left(\frac{x-b}{\sqrt{2}c}\right)\right]$$

Cumulative forms of the equations were needed to model the transitions from egg or to adult. For the larval and pupal stages, the distinction between fitting a Gaussian or modified Gaussian equation usually depended on length of time in stage. Because longer lasting stages often had a plateau when all individuals were in the same stage between transitions, the modified Gaussian relationship was more appropriate. Fitting these relationships provided evidence for the mathematical distribution of individuals during stage transitions.

A different regression procedure was needed to determine the duration of individual stages (50% of L1 to L2, for example). Various approaches could be used, for example, determining the time from the peak of one stage to the peak of the next. However, we used the time between 50% transition into a stage and 50% transition out of a stage. We made this choice because we can determine a standard error in the 50% transition point, which is not always possible when determining peaks. Determining the 50% transition point itself is straightforward through the use of a probit model, with the probit choices of being in the first stage or the next. Through probit modeling, it is possible to determine any desired % transition and the associated variation. Probit models were constructed with Prism 6.02.

For all regression analyses, the data were examined closely to determine their propriety for inclusion in the analysis. In a few instances, individuals were sampled for extraordinarily extended durations. These were treated as outliers and excluded from the analysis. Details on all data used are included in Appendix A.

### 2.5. Degree Days

Degree-day requirements were calculated with a combination of conventional regression analyses and iterative analyses to ensure that the resulting degree-day models reflected only the linear portion of the insect development curve. The outline of these procedures is the following:Determine the stage transitions by fitting cumulative Gaussian curves to the proportion of insects entering the new stage vs. time for each temperature (curves were calculated for L1, L2, L2f, L3m, P, and A). Only data for the first portion of each curve (0–100%) were included in the regression, which reflects the stage transition;Calculate the 50% transition point from the cumulative Gaussian curve for each stage and temperature combination;With data from 2, determine the time in stage by subtraction between 50% transition points;Express development times in days (rather than in hours as data were initially determined) and calculate 1/days for each time to transition and stage duration;Using linear regression, estimate the relationship between development rate (1/days to transition or stage) vs. temperature to determine the slope and x-intercept. Each resulting regression was run test to identify non-linearity, and where non-linearity was indicated, points were excluded from the regression until any non-linearity was eliminated. Primarily, non-linearity was associated with low and high temperatures (as expected) and indicated in development graphs. The regression of 1/days vs. temperature is conventional in degree-day determination, but the use of run testing to identify non-linear points in the regression has not been. To the best of our knowledge, this approach was first used in [14] to ensure that assumptions underlying degree-day analysis were met;From the resulting linear regressions, the x-intercept represents the developmental minimum, and 1/slope represents the accumulated degree days required for an event (stage transition or stage duration) [19]. Although this point usually represents the end of most degree-day determinations, we recognized that it is still possible at this point to have included data in the linear regressions that are not properly part of the linear portion of the development curve. Consequently, we performed additional calculations and corrections to determine the validity of our degree-day models;Using regression results, we calculated degree-day accumulations for each experimentally determined combination of temperature and time of transition or stage duration. We then performed a linear regression of these data and evaluated the resulting lines for linearity and slope. To meet the core assumption of degree-day models, a regression of degree-day accumulations must be linear and have no slope. Where our results did not meet these requirements, we removed points (again, at high and low temperatures) and recalculated both the 1/days regression and the accumulated degree-day regressions (steps 5–7). We repeated this process until we arrived at linear relationships meeting all degree-day assumptions and noted the range of temperatures for which the resulting equation was valid.

In the conventional use of degree days (e.g., [21]), multiple methods to ensure that only linear development data are used in determining degree-day models are not undertaken. Presumably, this omission has occurred because it is well recognized that degree-day models use assumptions of linearity to describe what is known to be a curvilinear relationship, so approaches for improving accuracy have focused on curvilinear model development [3,22] rather than on improving linear degree day accuracy. Additionally, most conventional uses of degree days with insects involve modeling population-level phenomena, where other sources of error (particularly in temperature data) and resolution (of days) are such that more precision in how degree-day models are developed may not be warranted. In contrast, with the forensic use of degree-day models, the potential inaccuracy associated with including non-linear data in the calculation model introduces a systematic error that could easily be significant in using degree days for estimating postmortem intervals [3].

## 3. Results and Discussion

There was no egg eclosion at 7.5 °C and no maturation past L1 at 10.0 °C, so those data are not reported here. However, there is evidence that the biological minimum for *P. regina* is between 10.0 °C and 12.5 °C, which agrees with development data reported by [13,14].

Large variation was observed during the L3m and pupation stages, regardless of temperature, with the variation being the greatest at 12.5 °C through to 20.0 °C. These stages are also the longest life stages (by proportion) (Table 2). In six of the 10 temperatures, we were not able to record 100% of the L3m to pupa transition because the variation in the transition times exceeded the a priori sampling time estimates (Figure 1).

*Phormia regina* did not display increased mortality associated with the upper temperature of 32.5 °C compared with *L. sericata*, where approximately 50% of the samples did not make it to adulthood [4]. Although the two species have differing development at extreme (or approaching extreme) temperatures, generally, their development follows very similar patterns: all life stages are normally distributed, with large transition periods between stages, and variation is highest during the L3m and pupal stages. For both species (and probably most blow flies), the variation seen during the later life stages could be associated with maggot physiology and not necessarily temperature. The two stages immediately preceding adulthood are primarily spent storing fat bodies and finding suitable places to metamorphose, activities that are not directly dependent on external temperatures.

*Phormia regina*, at the research time, had been in a colony of over 100 generations; however, this is unlikely to have led to reduced genetic variation based on understandings of dipteran colony genetics from other fly and calliphorid species (S. Skoda pers. comm. [23,24]). Ephemeral resource dependence for life cycle completion is a game of unpredictability, making the skill of survival over a large range of environments an important method of survival. Because blow flies are dependent on carrion for a portion of their life cycle, it would not be surprising to see similar inherent variation in most blow fly species.

The relationship between development rates and temperature by stage can be seen in Figure 1. The egg stage is highly dependent on temperature, as shown by the steep slope. Based on the slope, the pupal stage is the least dependent on temperature. Figure 2 tests the linearity assumption of the accumulated degree days (ADDs) by temperature and life stage. Points at extreme temperatures will not meet the assumption of curvilinearity. Therefore, those temperatures cannot be used in ADD calculations. The assumption of linearity is met for all life stages but not all temperatures. L3m and the pupal stage have the shortest temperature range, from 17.5 °C to 32.5 °C, although curvilinear data points can be observed on both extremes of the remaining life stages, highlighting the importance of curvilinear models. Curvilinear models can incorporate both the linear and non-linear portions of development, making more accurate predictions possible at extreme temperatures, where development differs from the linear approximation. The remaining life stages range from 12.5 °C to 32.5 °C (Table 3). Degree-day values appear in Table 4.

In comparing *P. regina* and *L. sericata* [4], similarities between the two species and the general agreement of the linear development data to other incomplete data sets suggest overall patterns in blow fly development, which, if true, has implications for sampling protocols and future development models that include the curvilinear portions that currently do not work with ADD calculations. Sampling the largest individuals is considered a “common practice” when collecting samples at a death scene, with the assumption that the largest individuals are the oldest. Richards and Villet [17] argue that an investigator should target the apparently most developed speci-mens at the scene. Stage transitions also are integral to generating realistic development data, and both *L. sericata* and *P. regina* spend much more time in mixed-aged populations than in non-mixed populations. This means that sampling only the largest larvae or only those that look like they have been on the carcass the longest is not a true representation of the age structure present, nor does it lend much statistical credibility to any results generated from the ‘largest is oldest’ sampling scheme. No method has been proposed for estimating carrion insect age from a set of specimens from a corpse. Age estimation is done one specimen at a time. Although there are no discipline-wide sampling protocols, the establishment of them would greatly reduce some of the error-related issues.

This model of *P. regina* development offers a baseline for considering other potential factors that may influence development, including geographic location, population density, food type and availability, and photoperiod. In the future, having an accurate understanding of blow fly development will consist of models that show the nature of stage transitions and incorporate curvilinearity at extreme developmental temperatures. Although these models do not currently exist, comprehensive data sets like the ones presented here make it more likely they will soon be a reality.

## Figures and Tables

**Figure 1 insects-15-00550-f001:**
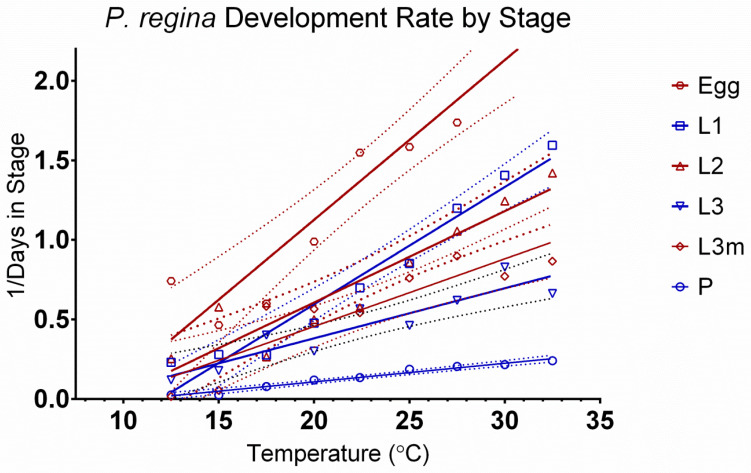
Development rate of *Phormia regina* by stage. Life stages of egg through pupation are represented by 10.0 to 32.5 °C. Confidence intervals are represented by dotted lines.

**Figure 2 insects-15-00550-f002:**
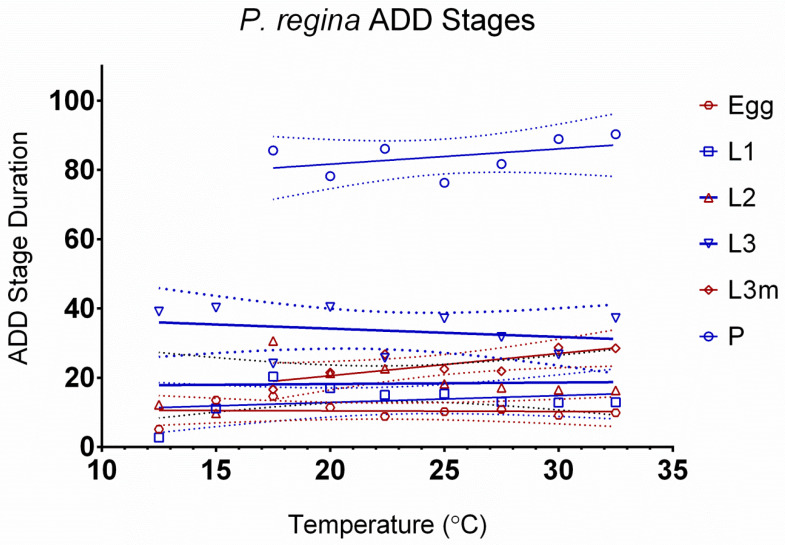
Accumulated degree-day stage durations of *Phormia regina*. Life stages of egg through pupation are represented by 10.0 to 32.5 °C. Confidence intervals are represented by dotted lines.

**Table 1 insects-15-00550-t001:** Methods comparison between the six commonly cited development papers for *Phormia regina*. Stages: E = egg, L1 = 1st instar, L2 = 2nd instar, L3f = 3rd instar feeding, L3m = 3rd instar postfeeding, migratory, P = pupa, and A = adult.

Reference	Locality	Temp.	Analysis	Larval Diet	Stages	L:D Cycle	Replications	Total Maggots/ Sample	Sample Times
[10]	WA U.S.	27.6	Mode	Beef liver	E, L1, L2, L3f, L3m, P	Constant	Undef.	Undef.	Undef.
[8]	IL, U.S.	19, 22, 29, 35	Minimum	Ground beef	E, L1, L2, L3f, L3m, P	Undef.	Undef.	Undef.	Undef.
[11]	BC, Canada	16.1, 23.0	Minimum and Maximum	Beef liver	E, L1, L2, L3f, L3m, P	Undef.	3	20, returned to jar	Eggs-1 to 2 h L1, L2- 3 to 4 times/day Later stages-2 to 3 times/day
[12]	FL, USA	10, 15, 20, 25, 30, 35, 40	Mean, mode	Lean pork	E, L1, L2, L3f, L3m, P	Egg-constant Larvae-12:12 Pupae- constant	Egg-3 Larvae-6, with 3 subsamples in each	6 from each subsample (= 108/sample)	Egg-30 min Larvae-12 h Onset of adult emergence-every 30 min
[13]	NE, U.S.	12, 14, 20, 26, 32 (2001) 12, 15, 20, 25, 30 (2004)	Mean	Ground beef, beef liver	E to P, E to A	16:8 (2001) 24:0 (2004)	26, 32-4 8, 10, 14, 20-2 12-1 (2001) All temps-4 (2004)	Undef.	12 h

**Table 2 insects-15-00550-t002:** Sample times for *Phormia regina* were calculated by converting the minimum and maximum data reported in Kamal (1958) into accumulated degree hours (ADHs). The ADHs were calculated for each life stage and sampling temperature, converted back into hours and divided into 5 equal sample times.

	Temperature (°C)										
Life Stage	7.5	10.0	12.5	15.0	17.5	20.0	22.5	25.0	27.5	30.0	32.5
Egg–1st	16	16	16	8	5	4	3	3	2	2	2
1st–2nd	44	44	44	22	15	11	9	7	6	6	5
2nd–3f	63	63	63	31	21	16	13	10	9	8	7
3f–3m	111	111	111	55	37	28	22	18	16	14	12
3m–Pupal	281	281	281	141	94	70	56	47	40	35	31
Pupal–Adult	441	441	441	221	147	110	88	74	63	55	49

**Table 3 insects-15-00550-t003:** Percent of *Phormia regina* individuals in stage by temperature. Temperatures reported were measured inside growth chambers.

	% Time in Stage					
Temp	Egg	L1	L2	L3f	L3m	P
10.4	N/A	N/A	N/A	N/A	N/A	N/A
12.7	4.8	19.6	14.5	3.3	15.9	41.8
15.1	6.4	9.6	7.9	22.3	2.3	51.5
17.5	6.4	14.0	14.6	9.6	6.6	48.8
20.1	5.4	11.3	10.8	17.8	9.5	45.3
22.5	4.4	9.6	11.7	11.9	12.4	49.9
25.0	5.3	9.9	9.9	18.3	11.1	45.4
27.5	5.8	8.4	9.5	16.2	11.2	49.0
30.0	4.7	7.8	8.9	13.3	14.3	51.0
32.5	4.9	7.3	8.2	17.6	13.5	48.5
Mean	5.3	10.8	10.7	14.5	10.8	47.9

**Table 4 insects-15-00550-t004:** Developmental data and linear regression results. Linear regression results (from Graph Pad Prism) for *Phormia regina*, with excluded (non-linear) points indicated by empty cells. Accumulated degree days (ADDs) were indicated by the slope of the regression line, and the developmental minimum was indicated by the x-intercept value [19]. For comparison, regression-based ADD was compared to mean ADD calculated across temperatures. The range of the linear regression indicates the temperature limits at which the assumption of linearity between temperature and development is valid.

Temp (mean)	Transition ADD by 1/Day						Stage ADD by 1/Days					
	E-L1	E-L2	E-L3f	E-L3m	E-P	E-A	Egg	L1	L2	L3f	L3m	P
10.2												
12.5												
15.0	8.9	13.7	21.3	58.8			8.9	6.0	7.9			
17.5	10.0	24.6	42.8	65.6	94.0	191.2	10.0	14.7	18.9	23.3	26.2	89.2
20.0	8.6	22.2	37.0	69.0	97.8	184.1	8.6	13.8	15.0	39.3	31.4	80.5
22.4	7.0	19.7	36.6	58.8	88.7	181.5	7.0	12.7	17.2	25.1	37.1	88.2
25.0	8.5	21.9	36.2	67.5	93.9	175.3	8.5	13.5	14.5	36.4	29.9	77.8
27.5	9.2	20.6	34.6	62.1	86.7	172.8	9.2	11.7	14.2	31.2	28.1	83.0
30.0	7.9	19.5	33.4	57.1	86.8	179.9	7.9	11.7	14.0	26.3	35.9	90.2
32.5	8.8	20.5	34.5	67.3			8.8	11.9	14.1			91.5
	**Linear Regression Results**											
Dev Min	11.5	12.9	12.8	11.9	10.5	10.2	11.5	13.5	12.5	8.2	2.3	10.5
ADD	8.4	20.4	34.7	62.9	91.4	181.9	8.4	12.1	14.4	29.3	30.2	84.9
r^2^	0.96	0.99	0.99	0.97	0.98	0.99	0.96	0.98	0.97	0.74	0.72	0.98
n	8	8	8	8	6	6	8	8	8	6	6	7
ADD Range min (°C)	15.0	15.0	15.0	15.0	15.0	15.0	15.0	15.0	15.0	17.5	17.5	15.0
ADD Range max (°C)	32.5	32.5	32.5	32.5	32.5	32.5	32.5	32.5	32.5	30.0	27.5	32.5
Calculated ADD mean	8.6	20.3	34.6	63.3	91.3	180.8	8.6	12.0	14.5	30.2	31.4	85.8
SE	0.8	2.9	5.7	4.4	4.2	6.0	0.8	2.5	3.0	5.9	4.0	4.9
Regression ADD	8	20	35	63	91	182	8	12	14	29	30	85
% deviation (calculated vs. regression ADD)	2.1%	−0.5%	−0.4%	0.6%	−0.1%	−0.6%	2.1%	−1.0%	0.7%	3.1%	4.2%	1.0%

## Data Availability

Data used in this study are provided in Appendix A, specifically the excel file “P regina development data.xlsx”.

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
