# Peer review of "Development Modeling of Phormia regina (Diptera: Calliphoridae)"

_insects, 2024, doi:10.3390/insects15070550_

Round 1

Reviewer 1 Report

Comments and Suggestions for Authors

„development rate is approximately a linear function of temperature“ is incorrect, because it assumes endless increase with high temperatures and negative values below developmental threshold. More appropriate would be „linear fractional function“, or „linear function in intermediate temperatures“.

Tab.1 What means „E“? Unexplained throughout text

„To meet the core assumption of degree day models, a regression of degree day accumulations must be linear and have no slope“. Nonlinear (SDR) models and degree days (SET models) are essentially identical, only SET is a special case of SDR under presumption of (fractional) linearity of SDR curve.

„the biological minimum for P. regina is between 10.0° C and 12.5° C“.  What is biological minimum? You mean lower temperature threshold for development? Or lower lethal t.t.? What about diapause in winter: are low temperatures above or below “biological minimum”?

Several times (e.g. line 312) author citation and not numbers used in text.

Blank lines 337-340 (??)

In „References“: why numbers are doubled?

Unify n- and m-dash (e.g. in references between paginations).

Lines 363 and 377- long dash between words.

It would be great to consult also the following: Development of Phormia regina at seven constant temperatures for minimum postmortem interval estimation Zhang, RN; Hu, GW; (...); Tao, LY, or Laboratory Development and Field Validation of Phormia regina (Diptera: Calliphoridae). Núñez-Vázquez, C; Tomberlin, JK; (...); García-Martínez,

I recommend to consult also some basic literature concerning development time modelling from the field of plant protection.

Maybe, authors should mention some other factors influencing development time (and probably also their results), for example: food availability, population density, photoperiod, etc.

Author Response

Tab.1 What means „E“? Unexplained throughout text
abbreviations for stages, key added to table

„To meet the core assumption of degree day models, a regression of degree day accumulations must be linear and have no slope“. Nonlinear (SDR) models and degree days (SET models) are essentially identical, only SET is a special case of SDR under presumption of (fractional) linearity of SDR curve.

„the biological minimum for P. regina is between 10.0° C and 12.5° C“.  What is biological minimum? You mean lower temperature threshold for development? Or lower lethal t.t.? What about diapause in winter: are low temperatures above or below “biological minimum”?
The ‘biological minimum” is the temperature below which development does not occur. The lower developmental threshold, the value used in calculating degree days, is NOT the same as the biological minimum, rather it is a value derived from the linear regression for the degree-day model and has no biological meaning. Text was added clarifying this point

Several times (e.g. line 312) author citation and not numbers used in text.
corrected

Blank lines 337-340 (??)
corrected

In „References“: why numbers are doubled?
? we have no doubling of number in our copy of the paper

Unify n- and m-dash (e.g. in references between paginations).
corrected

Lines 363 and 377- long dash between words.
corrected

It would be great to consult also the following: Development of Phormia regina at seven constant temperatures for minimum postmortem interval estimation Zhang, RN; Hu, GW; (...); Tao, LY, or Laboratory Development and Field Validation of Phormia regina (Diptera: Calliphoridae). Núñez-Vázquez, C; Tomberlin, JK; (...); García-Martínez,
references added

I recommend to consult also some basic literature concerning development time modelling from the field of plant protection.

Maybe, authors should mention some other factors influencing development time (and probably also their results), for example: food availability, population density, photoperiod, etc.
added

Reviewer 2 Report

Comments and Suggestions for Authors

The authors investigated the influence of temperature on the rate of pre-adult development of a forensically important blow fly Phormia regina. Although this influence was studied by several researches, the present study differed by a very detailed statistical analysis of a very carefully obtained data (large sample sizes obtained at a large range of temperatures with a small interval between the tested temperatures). Thus, considering the importance of the final aim of the study (an exact estimation of the post-mortem interval) I think that the manuscript can be published, although before publication a number of improvements and corrections should be made (see my comments below).

Lines 5-8: Please, include the country (USA) in the affiliations of the authors.

Line 22 etc. The terms “pupation” (lines 22, 110, 138, etc) and “pupariation” (lines 45 and 155) are used, as far as I understand, for the same process. Please, use the same term over the whole text and tables or clearly explain the difference between the two terms.

Lines 24,47, etc.: Again, the authors use different terms for the same parameter,  a minimum temperature at which development is possible: “the biological minimum” (lines 24, 262, 265), “a minimum development (developmental) temperature” (lines 47 and 48), and “the developmental minimum” (lines 231, 311).

First, I would strongly recommend using of one and the same term over the whole paper. Second, although this is not mandatory, I would suggest the term "the lower developmental threshold" that is much more common in scientific literature.

Lines 25-26 and 276-277: I can’t find in the present manuscript any data on Phormia regina mortality at different temperatures. Please, either include these data in the manuscript or delete these statements concerning mortality because all conclusions should be supported by the data.  

Line 30: I would add “temperature; development; blow fly; Phormia regina” to the list of key words.

Lines 49-50: Of course, “even temperature intervals” looks nice in any graph, but in fact linear regression can be calculated using the data obtained at unequal intervals, too (the exactness of the prediction increases with the number of tested temperatures within the optimal range). Thus, I would not agree that “Such models require even temperature intervals”.

Line 95: “At research time...”. Please, indicate here the year when the research was conducted to estimate the age of the laboratory population used.

Lines 97-99: The second quotes are absent in this sentence. 

Lines 95-99 and 286-287.

From one hand, the authors stated that “colonies had achieved 100 generations without addition of new flies to reduce genetic variation within the colony. The intent was to obtain genetic homogeneity among test subjects” (lines 95-99). On the other hand, they concluded that “Phormia regina, at research time, had been in colony over 100 generations, however, this is unlikely to have led to reduced genetic variation” (lines 286-287). It seems to be a contradiction. What of the two statements is true? Please, explain it.

Line 118: Well, and what humidity (RH percentage) was maintained by this recirculating air system in your study? Please, include this information in the Methods.

Line 133: It is a significant drawback of the study that the maximum studied temperature was 32.5 C. This temperature did not cause an increase in mortality (lines 276-277) suggesting that it was substantially lower than the higher developmental threshold. Normally, such studies should be preceded by pilot experiments aimed to estimate the tolerance temperature range of the studied species. However, it seems that the authors used the data for the tolerance range of an earlier studied L. sericata (lines 277-278). Please, consider this problem in the Methods and Discussion.

Lines 221-222: Add or delete one bracket in this sentence.

Table 3: The sum of the mean percents (the bottom line of the table) is 104.8% which looks a bit strange because, as far as I understand, the sum of the six rounding errors can’t exceed 0.6%. Possibly, something is wrong in your table or in my calculations? Please, either correct or explain it.

Lines 281-285: Please, provide references for these statements.

Lines 343-381: The serial numbers of references are duplicated. Please, correct it.

Author Response

Lines 5-8: Please, include the country (USA) in the affiliations of the authors.
done

Line 22 etc. The terms “pupation” (lines 22, 110, 138, etc) and “pupariation” (lines 45 and 155) are used, as far as I understand, for the same process. Please, use the same term over the whole text and tables or clearly explain the difference between the two terms.
done

Lines 24,47, etc.: Again, the authors use different terms for the same parameter,  a minimum temperature at which development is possible: “the biological minimum” (lines 24, 262, 265), “a minimum development (developmental) temperature” (lines 47 and 48), and “the developmental minimum” (lines 231, 311).
as discussed in response to reviewer 1, the biological minimum and the minimum developmental temperature are distinctly different concepts. The biological minimum the temperature below which development will not occur; the minimum developmental temperature is a value derived from the linear regression in a degree-day model and has no biological significance. Text has been added to clarify this point. (See Higley and Haskell 2010 for more explanation).

First, I would strongly recommend using of one and the same term over the whole paper. Second, although this is not mandatory, I would suggest the term "the lower developmental threshold" that is much more common in scientific literature.

Lines 25-26 and 276-277: I can’t find in the present manuscript any data on Phormia regina mortality at different temperatures. Please, either include these data in the manuscript or delete these statements concerning mortality because all conclusions should be supported by the data.  
Although we have data on mortality by stage for all our experiments, we think including these data would distract from the main points of the paper. Moreover, we only experienced high mortality (100%) at the lowest temperatures. I believe this is the only mortality statement we have included in the manuscript.

Line 30: I would add “temperature; development; blow fly; Phormia regina” to the list of key words.
key words added, except for those already occurring in the title

Lines 49-50: Of course, “even temperature intervals” looks nice in any graph, but in fact linear regression can be calculated using the data obtained at unequal intervals, too (the exactness of the prediction increases with the number of tested temperatures within the optimal range). Thus, I would not agree that “Such models require even temperature intervals”.
We disagree with the reviewer on this point. While it is common to use uneven intervals in linear regressions (certainly we’ve done so on occasion), the use of uneven intervals introduces potential bias in the regression. This is clearly indicated with regressions having one outlying point which alters the regression relationship in a disproportionate fashion – the single point has more influence on the regression than the other (clustered) points. In most instances, where x values intervals are not widely disparate, the bias is negligible, however, we have seen, in the literature, a number of instances where single, low temperature points are included with other temperature points clustered at higher temperatures. In these instances bias is significant.

Line 95: “At research time...”. Please, indicate here the year when the research was conducted to estimate the age of the laboratory population used.
added

Lines 97-99: The second quotes are absent in this sentence. 
corrected

Lines 95-99 and 286-287.

From one hand, the authors stated that “colonies had achieved 100 generations without addition of new flies to reduce genetic variation within the colony. The intent was to obtain genetic homogeneity among test subjects” (lines 95-99). On the other hand, they concluded that “Phormia regina, at research time, had been in colony over 100 generations, however, this is unlikely to have led to reduced genetic variation” (lines 286-287). It seems to be a contradiction. What of the two statements is true? Please, explain it.
These statements say the same thing. “Genetic homogeneity” means uniformity in genetics or a lack of variation. The second sentence expresses the same notion: that over 100 generations we are unlikely to have high genetic variation. The purpose of seeking reduced genetic variation is to avoid high levels of noise in development data associated with genetic variation.

Line 118: Well, and what humidity (RH percentage) was maintained by this recirculating air system in your study? Please, include this information in the Methods.
added

Line 133: It is a significant drawback of the study that the maximum studied temperature was 32.5 C. This temperature did not cause an increase in mortality (lines 276-277) suggesting that it was substantially lower than the higher developmental threshold. Normally, such studies should be preceded by pilot experiments aimed to estimate the tolerance temperature range of the studied species. However, it seems that the authors used the data for the tolerance range of an earlier studied L. sericata (lines 277-278). Please, consider this problem in the Methods and Discussion.
Actually, we disagree here regarding the importance of including temperatures above 32.5 C. Previous research (on P. regina [12]) indicated that above 32.5 C the developmental curve would be significantly non-linear and therefore unusable for developing degree-days. We added text to make this point more clear.

Lines 221-222: Add or delete one bracket in this sentence.
fixed

Table 3: The sum of the mean percents (the bottom line of the table) is 104.8% which looks a bit strange because, as far as I understand, the sum of the six rounding errors can’t exceed 0.6%. Possibly, something is wrong in your table or in my calculations? Please, either correct or explain it.
Many thanks – there was an error in the original table. We went back to the original data and corrected this. Rows and means total to 100% as they should.

Lines 281-285: Please, provide references for these statements.
done

Lines 343-381: The serial numbers of references are duplicated. Please, correct it.
This error does not appear on our formatted draft.

Round 2

Reviewer 2 Report

Comments and Suggestions for Authors

The authors addressed all my comments and either accepted them or provided detailed explanations. The manuscript was substantially improved. I think that now it can be published.